# Mendelian randomization integrating GWAS and eQTL data reveals genetic determinants of complex and clinical traits

Eleonora Porcu [1,2], Sina Rüeger [2,3], Kaido Lepik [3,4], eQTLGen Consortium, BIOS Consortium, Federico A. Santoni[5], Alexandre Reymond[1,65] & Zoltán Kutalik[2,3,65]

Genome-wide association studies (GWAS) have identified thousands of variants associated with complex traits, but their biological interpretation often remains unclear. Most of these variants overlap with expression QTLs, indicating their potential involvement in regulation of gene expression. Here, we propose a transcriptome-wide summary statistics-based Mendelian Randomization approach (TWMR) that uses multiple SNPs as instruments and multiple gene expression traits as exposures, simultaneously. Applied to 43 human phenotypes, it uncovers 3,913 putatively causal gene–trait associations, 36% of which have no genome-wide significant SNP nearby in previous GWAS. Using independent association summary statistics, we find that the majority of these loci were missed by GWAS due to power issues. Noteworthy among these links is educational attainment-associated *BSCL2*, known to carry mutations leading to a Mendelian form of encephalopathy. We also find pleiotropic causal effects suggestive of mechanistic connections. TWMR better accounts for pleiotropy and has the potential to identify biological mechanisms underlying complex traits.

[1] Center for Integrative Genomics, University of Lausanne, Lausanne, Switzerland. [2] Swiss Institute of Bioinformatics, Lausanne, Switzerland. [3] University Center for Primary Care and Public Health, University of Lausanne, Switzerland, Lausanne, Switzerland. [4] Institute of Computer Science, University of Tartu, Tartu, Estonia. [5] Endocrine, Diabetes, and Metabolism Service, CHUV and University of Lausanne, Lausanne, Switzerland. [65]These authors jointly supervised this work: Alexandre Reymond and Zoltán Kutalik. A full list of consortium members appears at the end of the paper. Correspondence and requests for materials should be addressed to E.P. (email: eleonora.porcu@unil.ch) or to Z.K. (email: zoltan.kutalik@unil.ch)

Genome-wide association studies (GWAS) have identified tens of thousands of common genetic variants associated with hundreds of complex traits[1]. The identification of causal genes using GWAS results is difficult, however, because this approach only highlights fine-mapped intervals of associated variants in linkage disequilibrium (LD) with the causal marker[2–4]. Linking the effect of SNPs to gene function is not straightforward without additional data, especially, as the majority of these trait-associated variants fall into non-coding regions of the genome[1] with no direct influence on protein structure or function.

It has been shown that trait-associated SNPs are three times more likely to be associated with gene expression, i.e., expression quantitative trait loci (eQTLs)[5–8]. Such significant enrichment suggests that many SNP-trait associations could act through gene expression (i.e., SNP → Gene expression → trait).

Transcriptome-wide association studies (TWAS) integrating GWAS and eQTLs data have been proposed to unravel gene–trait associations[7,9,10]. However, although these studies aim to identify genes whose (genetically predicted) expression is significantly associated to complex traits, they do not aim to estimate the strength of the causal effect and are unable to distinguish causation from horizontal pleiotropy (i.e., when a genetic variant influences multiple phenotypes independently). For this reason, we rather chose to apply a Mendelian randomization (MR) approach to estimate the causal effect of gene expression on complex traits.

A conventional MR analysis estimates the causal effect of a risk factor (exposure) on an outcome by using genetic variant(s) that are (directly) associated only with the risk factor as instrumental variables[11]. As most SNPs have small effects on phenotypes, increasing the number of instruments increases the statistical power[12]. If SNPs, exposure, and outcome are all measured in the same sample, the causal effect of the risk factor in the outcome can be estimated using a two-stage least squares[13] approach. However, large cohorts of this kind are rare rendering such approach heavily underpowered. It is often the case that the exposure and the outcome are available in different data sets, a situation for which two-sample MR methods have been developed[14]. The key advantage of using two-sample MR is that it only requires publicly available GWAS summary statistics[15]. In such an approach, only independent SNPs are considered and a fixed effects inverse–variance weighted (IVW) meta-analyses is used to estimate the causal effect[16,17]. As the IVW estimate is a weighted average of the effects from each SNP, if any of the SNPs shows horizontal pleiotropy then the causal effect estimate is biased. However, such pleiotropy introduces heterogeneity, which can be detected and SNPs contributing the most to the heterogeneity can be excluded[18–20], a good solution if the majority of the instruments are valid. Conversely, when some of the genetic variants in the analysis are not valid instruments, other MR approaches (i.e., MR-Egger[20], weighted median[21], or mode-based MR[22]) should be applied. Although such methods provide a more robust estimate of the causal effect, they have less power to detect causal association.

Pleiotropy could alternatively be tackled using multivariable MR. If a variant exhibits horizontal pleiotropy, but we know its association(s) to some mediators of its indirect effect to the outcome, those mediators could be included as additional exposures and one can perform a multivariable MR, which can mitigate bias by jointly estimating the causal effects of all exposures on the outcome[23,24].

Here, we adapt a multivariable MR method tailored to gene expression levels as exposure, termed TWMR (Transcriptome-Wide Mendelian Randomization), which integrates summary-level data from GWAS and eQTLs studies in a multivariable MR framework to estimate such causal effect of gene expression on complex human traits. Of note, throughout the manuscript the term "causal" is used for simplicity to mean "found to be causal by our MR method". A previous study[25] proposed a univariable, single instrument MR approach to identify genes whose expression levels are associated with a complex traits. However, it had limited means to distinguish causality from pleiotropy. As eQTLs are often shared between multiple genes[26], we find it beneficial to adapt a multiple-instrument, multiple-exposure MR approach for gene expression exposures. Our method only requires summary-level data (along with pair-wise SNP LD estimated from, e.g., the 1000 Genomes[27] or the UK10K reference panel[28]) allowing data integration from different studies.

The manuscript is organized as follows. First, we perform an extensive simulation study to confirm that under our model setting, the method controls type I error rate and achieves superior RMSE compared with standard approaches. We then apply our method to the largest publicly available GWAS summary statistics (based on sample sizes ranging from 20,883 to 339,224 individuals) and combine them with eQTL data from GTEx[29] and the eQTLGen Consortium ($n = 31,684$)[30] to provide an atlas of putatively functionally relevant genes for 43 complex human traits. As there are only sporadic examples of causal gene–disease links, we use the following proxies to gold standard gene–disease links and test whether the TWMR results are meaningful and confirm previous knowledge: (a) experimentally established causal links (e.g., *SORT1* with LDL in liver[31]); (b) gene–disease links based on the OMIM database; (c) genes falling into an association region identified by GWAS, but only in larger sample size. Finally, we carry out several follow-up analyses to make biological inferences.

## Results

**Overview of the approach**. MR relies on three assumptions about the instruments: (i) they must be sufficiently strongly associated with the exposure; (ii) they should not be associated with any confounder of the exposure–outcome relationship; and (iii) they should be associated with the outcome only through the exposure. Violation of any of these assumptions would lead to biased estimates of the causal effect and potential false positives[18]. In our view, one of the most difficult situations in an MR analysis is when there exists a heritable confounding factor of the exposure–trait relationship. All instruments for the exposure that act through this confounding factor will have proportional effect on the exposure and the outcome and will bias the causal effect estimates toward the ratio of the causal effect of the confounding factor on the outcome and the risk factor. Whereas in the past few years methodological developments have addressed pleiotropy issues[18,19,17], the situation described above was rarely addressed[32].

Recently, Zhu et al.[25] developed a colocalization method in a summary-based MR analyses framework to test whether the effects of genetic variants on a phenotype are mediated by gene expression. They designed a heterogeneity test (HEIDI) to identify pleiotropic SNPs that could bias the causal estimate. Using only one instrument renders the method less powerful and when small eQTL study is used there is often insufficient power for the heterogeneity test to flag up if the top eQTL is an outlier. Including other SNPs as instruments allows us to replace the third MR assumption with the weaker InSIDE (Instrument Strength Independent of Direct Effect) assumption. When the InSIDE assumption is violated, but entirely owing to another genes' expression being the confounders of the (primary gene) expression–trait correlation, including the confounder genes as an additional exposures in a multivariable MR can resolve the problem and yield unbiased causal effect estimates. For this

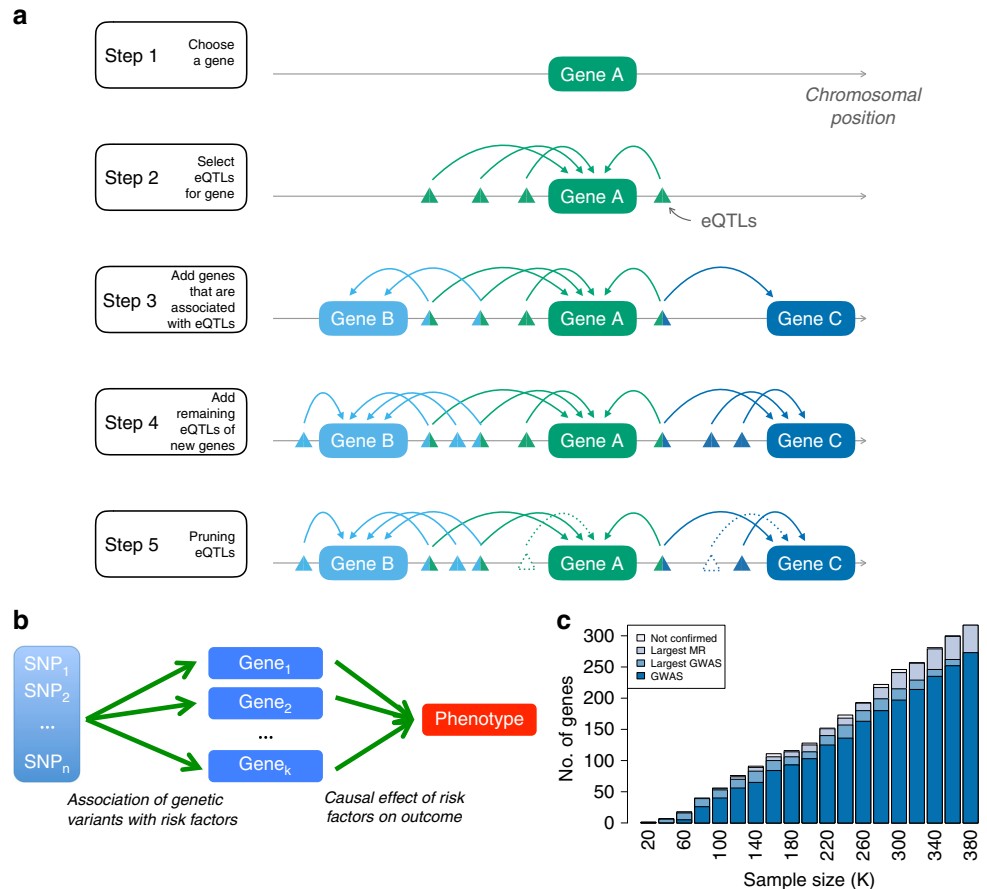

**Fig. 1** Overview of TWMR approach. **a** Workflow of the selection of SNPs and genes included in our analyses. For each gene (step 1), we consider all the independent eQTLs with $P_{eQTL} < 1.83 \times 10^{-5}$ (step 2) and then we included in the model all the genes for which those SNPs are eQTLs (step 3). In addition, we included in the model all the SNPs that are eQTLs only for these genes (step 4). To avoid multicollinearity issues, we pruned the SNPs keeping only independent SNPs (i.e., $r^2 < 0.1$) (step 5). **b** Schematic representation of TWMR to estimate the causal effect of multiple exposures (expression of genes) on a phenotype using multiple instrumental variables (SNPs). **c** Validation of genes found by TWMR. Bars represent the number of genes found to be causal for BMI when carrying out the TWMR analysis in different sized subsets of the UKBB. Among them, those confirmed by GWAS in the same sample (i.e., that fall within 500kb of a GW significant SNP) are marked in dark blue; those confirmed only when running GWAS in the full 380K UKBB samples in light blue; those confirmed only running TWMR in the full UKBB data set in gray, whereas the light gray bars on the top represent the number of genes found by TWMR, but not confirmed in the full UKBB data set

reason, we applied TWMR, a multivariable (multi-gene) multi-instrument MR approach[16,33] (Fig. 1a, b) that should specifically reduce the bias owing to pleiotropic effects.

For a set of $k$ genes, using inverse–variance weighted method for summary statistics[34,35], we estimated the multivariate causal effect of the expression levels for several genes at a locus on an outcome trait as

$$\hat{\alpha} = \left(\mathbf{E}'\mathbf{C}^{-1}\mathbf{E}\right)^{-1}\left(\mathbf{E}'\mathbf{C}^{-1}\mathbf{G}\right), \qquad (1)$$

where $\mathbf{E}$ is a $n \times k$ matrix that contains the univariate effect size of $n$ SNPs on $k$ gene expressions (these estimates come from an eQTL study); $\mathbf{G}$ is a vector of length $n$ that contains the univariate effect sizes of the same $n$ SNPs on the phenotype (these estimates come from publicly available GWAS summary statistics) and $\mathbf{C}$ is the pair-wise correlation (LD) matrix between $n$ SNPs (estimated from the UK10K panel).

To demonstrate the advantage of our multi-exposure approach, we performed simulation analyses in settings relevant for expression-disease causal analysis. In particular, we demonstrated that if a subset of SNPs affect more than one gene at a locus, the multi-gene approach provides a more precise estimation of the causal effect of the gene expression on the phenotypes: the root mean squared error (RMSE) in the multi-gene approach is more

than twofold lower than in the single-gene approach in all the simulations performed varying the degree of pleiotropy, number of genes, and SNPs included in the model. Furthermore, we showed that the multi-gene approach is more powerful (on average 1.3% power gain) and, more importantly, accurately controls type I error rate as opposed to the single-gene approach, which can easily reach 20% (at 5% nominal level) in case of pleiotropy (Supplementary Figs. 1–6).

**Applying TWMR to GWAS and eQTL summary statistics**. We applied TWMR to summary data from an eQTL meta-analyses performed in blood samples from >31 K individuals (eQTLGen Consortium[30]) and the largest publicly available GWAS data to assess causal associations between gene expression and 43 complex traits.

Note that the aim of our paper is not to confirm the validity of a well-established method in simulation studies, because such MR approaches have been extensively tested in various simulation setups[34,36]. Our focus is rather to demonstrate the utility of such approaches when applied to real data.

Data from eQTLGen Consortium contain association summary statistics for 10,023,016 SNPs and 19,251 genes. In our analyses we included only cis-eQTLs with FDR < 0.05 (corresponding to

$P_{eQTLs} < 1.829 \times 10^{-5}$), resulting in 3,699,824 eQTLs for 16,990 eGenes (i.e., genes whose expression levels are associated with at least one genetic variant). We only used cis-eQTLs data as trans-eQTLs typically have weaker effect size and less direct effect, hence are more prone to violate MR assumptions. In total, we found 2,369 genes putatively causally associated with at least one phenotype giving rise to 3,913 gene–trait associations ($P_{TWMR} < 3 \times 10^{-6} = 0.05/16,000$, where 16,000 corresponds to the number of genes tested for each phenotype) (Supplementary Fig. 7 and Supplementary Table 1 and Supplementary Data 1).

**Pleiotropic SNPs lead to biased causal effect estimates**. Using our approach in simulations, we show how single-gene analyses can lead to biased causal effect estimates (Supplementary Figs. 1–6 and 9). In the multi-gene approach, we condition the SNP-exposure effects on their corresponding effects on other genes but still, genetic variants can influence the outcome through other (non expression related) risk factors (horizontal pleiotropy). Under the assumption that the majority of SNPs in the region influence the outcome only through the exposures included in the model[22], SNPs violating the third MR assumption would significantly increase the Cochran's heterogeneity Q statistic (see Methods)[37]. Overall, we detected heterogeneity ($P_{HET} < 1 \times 10^{-4}$) for 2,017 of the 5,275 originally significant gene–trait associations. Out of these 2017 associations, 549 passed the heterogeneity test after removing SNPs showing pleiotropic effects. Furthermore, removing pleiotropic SNPs led to the identification of 106 additional associations, giving the final number of 3,913 robust associations (Supplementary Fig. 8).

**MR improves GWAS power to detect associated loci**. Our method, incorporating eQTL information into GWAS analyses, has the potential to increase the power of GWAS in identifying loci associated with complex traits.

Conventional gene-based tests, usually based on physical distance or LD rarely lead to the discovery of new loci[38,39]. With such approaches, identifying new loci missed by GWAS is very difficult because a limited set of (independent) associated variants in/near a gene is diluted by the large number of null SNPs. Hence, the combined association signal for genes is typically weaker than the strongest SNP in the region.

Furthermore, given that the majority of GWAS SNPs falls outside coding regions, they may be excluded from gene-centric analyses.

To evaluate the performance of our approach we conducted GWAS and TWMR analyses for BMI on several subsets of individuals from UKBiobank (UKBB). Using the full sample, we found 343 significant TWMR genes associated with BMI (Fig. 1c). Among these, 108 are >500kb away from any GW significant SNP, hence potentially representing new loci that were missed by the conventional GWAS owing to power issues. To assess if the additional genes implicate regions harboring truly associated SNPs, we carried out the GWAS and TWMR analysis for 18 subsets of the UKBB with increasing sample size (from 20,000 to 360,000). We observed that most of TWMR identified genes in the small subsets are confirmed, i.e., fall within 500kb vicinity of lead SNPs identified as GW significant in the GWAS using the full sample. For example, 16 genes found by TWMR and not found by GWAS (i.e., positioned >500kb away from any GW significant SNP) in the subset of 100K individuals overlap significantly (OR = 5.70, hypergeometric $P = 0.042$) with the 365 genes mapping within 500kb of the SNPs identified by the GWAS performed using the full data set. We observed that the portion of loci missed by GWAS decreases with increasing sample size, indicating a saturation effect. To support the consistency of our findings, >60% of TWMR genes identified in any of the 18 subsets of the full sample and missed by GWAS in the full data set show significant association in the TWMR analyses performed in the full data set.

**New trait-associated genes**. In total, we found 2,369 putative genes causally associated with at least one phenotype (3,913 gene–trait associations at $P_{TWMR} < 3 \times 10^{-06}$) (Supplementary Fig. 7, Supplementary Table 1 and Supplementary Data 1). Of these gene–trait associations, 36% (1,399) were not prioritized by previous GWASs, as no SNP reached genome-wide significance level within the gene ±500kb (Supplementary Fig. 10). Of note, 27% (1,068/3,913) were missed by GWAS even when using a less stringent threshold (i.e., $P_{GWAS} < 1 \times 10^{-6}$). For example we detected a causal effect of *BSCL2* ($P_{TWMR} = 1.89 \times 10^{-6}$) on educational attainment (Fig. 2a). This is the only gene showing a significant causal effect in that region and its positive link with educational attainment is consistent with the already known

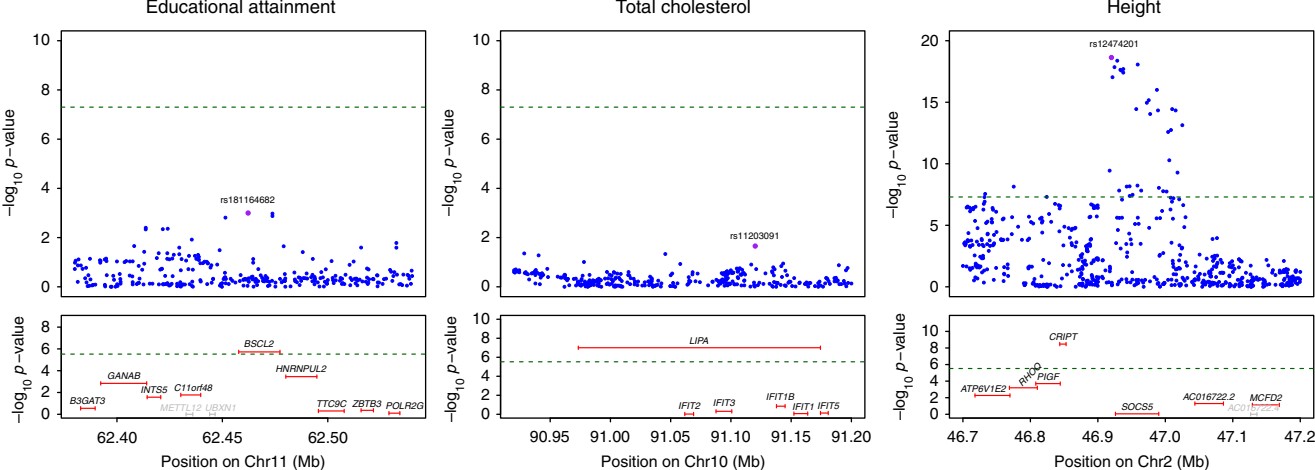

**Fig. 2** Regional association plot for GWAS and TWMR analyses. Top panels show single-SNP association strengths (y axis shows the −log₁₀(p value)) versus genomic position (on hg19/GRCh37 genomic build; x axis) around the most significant SNP, indicated with a purple dot. Lower panels illustrate gene causal association strengths (y axis shows the −log₁₀(p value)) versus genomic position. Genes highlighted in gray were not tested (i.e., they do not pass the heterogeneity test or not have significant eQTLs). Green dotted lines represent GWAS and TWMR significance thresholds. Note that for educational attainment and total cholesterol the most significant SNP did not reach GW significance (top), whereas TWMR pinpoints to putative causal genes (bottom)

involvement of this gene in type 2 congenital generalized lipodystrophy (BSCL) (OMIM:#615924)[40], which is frequently associated with some degree of intellectual impairment. In addition, among other regions missed by conventional GWAS, we show that the association of *LIPA* ($P_{TWMR} = 9.93 \times 10^{-8}$) (Fig. 2b) and *RAB23* ($P_{TWMR} = 2.71 \times 10^{-7}$) with total hypercholesterolemia[41] and height[42], respectively, is not restricted to rare coding variants.

Furthermore, we prioritized genes in regions already known to be associated with complex traits. For example, within the *SOCS5* locus, which harbors height-associated SNPs identified by Wood et al.[43] (top SNP: rs12474201 $P = 2.30 \times 10^{-19}$), our results suggest that *SOCS5* is not causal ($P_{TWMR} = 0.89$) and reveal that high expression of *CRIPT* ($P_{TWMR} = 3.40 \times 10^{-9}$) is causally linked to high stature (Fig. 2c). Rare mutations in *CRIPT* are known to be associated with short stature (OMIM:#615789)[44], making it the strongest candidate.

To test whether our putative causal genes are functionally relevant, we overlapped the genes significantly associated with height, educational attainment and total cholesterol with genes assembled from the Online Mendelian Inheritance in Man (OMIM) database involved in abnormal skeletal growth syndrome[43], cognitive impairment and hypercholesterolemia, respectively. Although we observed only a trend for enrichment for height (1.3-fold, hypergeometric $P > 0.05$) and total cholesterol (3.7-fold, hypergeometric $P > 0.05$), we found a significant enrichment for educational attainment (2.6-fold, hypergeometric $P = 0.005$) providing additional supporting evidence for our prioritized genes.

We also found 1,784 regions where TWMR identified only one putative causal gene. In two regions for educational attainment, our analysis pointed exclusively to *STRADA* ($P_{TWMR} = 1.32 \times 10^{-6}$) and *TBCE* ($P_{TWMR} = 2.90 \times 10^{-6}$), whose high and low expression, respectively, is associated with educational attainment. Of note, *STRADA* is already known to be associated with polyhydramnios, megalencephaly, and symptomatic epilepsy (OMIM:#611087)[45] and *TBCE* with hypoparathyroidism-retardation-dysmorphism syndrome (OMIM:#241410)[46]. In another example, for rheumatoid arthritis, *PSTPIP1* emerged as the only causal gene at its locus: its low expression is significantly associated with the risk of rheumatoid arthritis in the general population ($P_{TWMR} = 1.24 \times 10^{-7}$). *PSTPIP1* was associated with pyogenic sterile arthritis, pyoderma gangrenosum, and acne syndrome (OMIM:#604416)[47].

Disease-associated (non-synonymous) coding variants are likely to have direct impact on the protein sequence, in which case gene expression levels may be less important. To explore this hypothesis, we looked up the causal association of 44 testable genes harboring at least one coding variant associated with height[48]. Only five out of the 44 genes showed significant causal effect on height (OR = (5/44)/((374−5)/(13,849−44)) = 4.25 where 13,849 is the total number of testable genes for height and 374 is the number of TWMR-significant genes) and hypergeometric $P = 0.0047$). That enrichment is lower than the enrichment we observed overlapping the genes TWMR-significant with the 1,088 genes mapping within 100kb of the SNPs identified by the GWAS (OR = (280/1088)/(94/12761) = 34.9 and $P = 3.72 \times 10^{-281}$). This suggests that when the association is driven by coding variant and not by gene expression, our MR approach correctly does not point to any causal gene.

**Closest genes are often not causal**. Recent studies suggested that the gene closest to the GWAS top hits is often not the causal one[25,31,49]. Consistent with these findings, among the 1,125 TWMR-significant regions harboring at least one genome-wide significant SNP, we found that 71% of the genes closest to the top

SNP in the region do not show any significant association with the phenotype ($P_{TWMR} > 3 \times 10^{-6}$) (Supplementary Fig. 11). One of the numerous such examples is the significant causal association between educational attainment and *ERCC8* ($P_{TWMR} = 1.05 \times 10^{-13}$), a gene previously linked with the monogenic Cockayne Syndrome A (OMIM#216400)[50] in the *ELOVL7* region (top SNP: rs61160187, $P = 5.93 \times 10^{-13}$).

**Genes with pleiotropic effect**. Genes often play a role in numerous and independent biological processes, leading to different outcomes. We thus investigated the degree of pleiotropy and identified 848 genes associated with multiple phenotypes (Supplementary Fig. 7). One such pleiotropic gene is *GSDMB*: its low expression is associated with Crohn's disease (CD) ($P_{TWMR} = 2.89 \times 10^{-11}$), inflammatory bowel disease ($P_{TWMR} = 4.54 \times 10^{-11}$), rheumatoid arthritis (RA) ($P_{TWMR} = 2.91 \times 10^{-15}$), ulcerative colitis (UC) ($P_{TWMR} = 2.70 \times 10^{-08}$), high-density lipoprotein cholesterol ($P_{TWMR} = 4.27 \times 10^{-9}$), lymphocytes ($P_{TWMR} = 3.41 \times 10^{-23}$) and mean platelet volume ($P_{TWMR} = 2.68 \times 10^{-7}$). This result pinpoints shared effects across CD, UC, and RA. Of note, the susceptibility allele of the rs2872507 locus was already known to be associated with the reduced expression of *GSDMB* in intestinal biopsies from patients with IBD[51].

We also found significant association between *COPG1* and lipid traits: high-density lipoprotein cholesterol ($P_{TWMR} = 1.64 \times 10^{-6}$), low-density lipoprotein cholesterol ($P_{TWMR} = 7.88 \times 10^{-11}$), total cholesterol ($P_{TWMR} = 1.51 \times 10^{-9}$), and triglycerides ($P_{TWMR} = 8.85 \times 10^{-10}$). Of note, we observed significant causal associations, despite the fact that this locus was missed by the GWASs we used in our TWMR analysis for the four traits (Fig. 3). Supporting our findings, this gene is involved in lipid homeostasis.

**Trait correlation**. Exploring the relationships among complex traits and diseases can provide useful etiological insights and help prioritize likely causal relationships. A cross-trait LD Score regression method[52] was used to evaluate the genome-wide genetic correlation between complex traits. To possibly understand the biological mechanism of the shared genetic architecture we estimated the proportion of such genetic correlation channeled through the transcriptome program. For this, we computed the correlation ($\hat{\rho}_E$) between the causal effect estimates of the gene expression (or equivalently the $Z$ scores from our TWMR analysis) across a subset of 2,974 independent genes (including those that were not significant for any trait; see Methods). Among the 903 pairs of traits, we found several significant correlations in line with previous epidemiological observations (Fig. 4a). For example, for age at menarche we observed negative correlation with BMI ($\hat{\rho}_E = -0.20$ FDR $= 2.97 \times 10^{-20}$)[53,54]. As expected, we observed a negative correlation ($\hat{\rho}_E = -0.10$, $P_{FDR} = 2.20 \times 10^{-5}$) between coronary artery disease and educational attainment[55]. Out of the 43 traits included in our analyses, 17 were included in a previous study reporting genetic correlation between traits estimated by LD score regression ($\hat{\rho}_G$)[52]. Comparing the expression ($\hat{\rho}_E$) with genetic correlation ($\hat{\rho}_G$) estimates for 136 common pairs of traits, we found a remarkable concordance between the two estimates ($r = 0.84$). Of note, the expression correlation seems to be 43% of the genetic correlation on average. Although the genetic correlation estimate having smaller variance may explain part of this attenuation, we think that the main reason is that about half of the observed genetic correlation propagates to gene–expression level in whole blood (Fig. 4b). In particular we observed 33 pairs of traits showing significance for $\hat{\rho}_E$ and $\hat{\rho}_G$, whereas four were significant only for $\hat{\rho}_E$, 9 only for $\hat{\rho}_G$ and 90 not significant for either. Among the significant correlations not

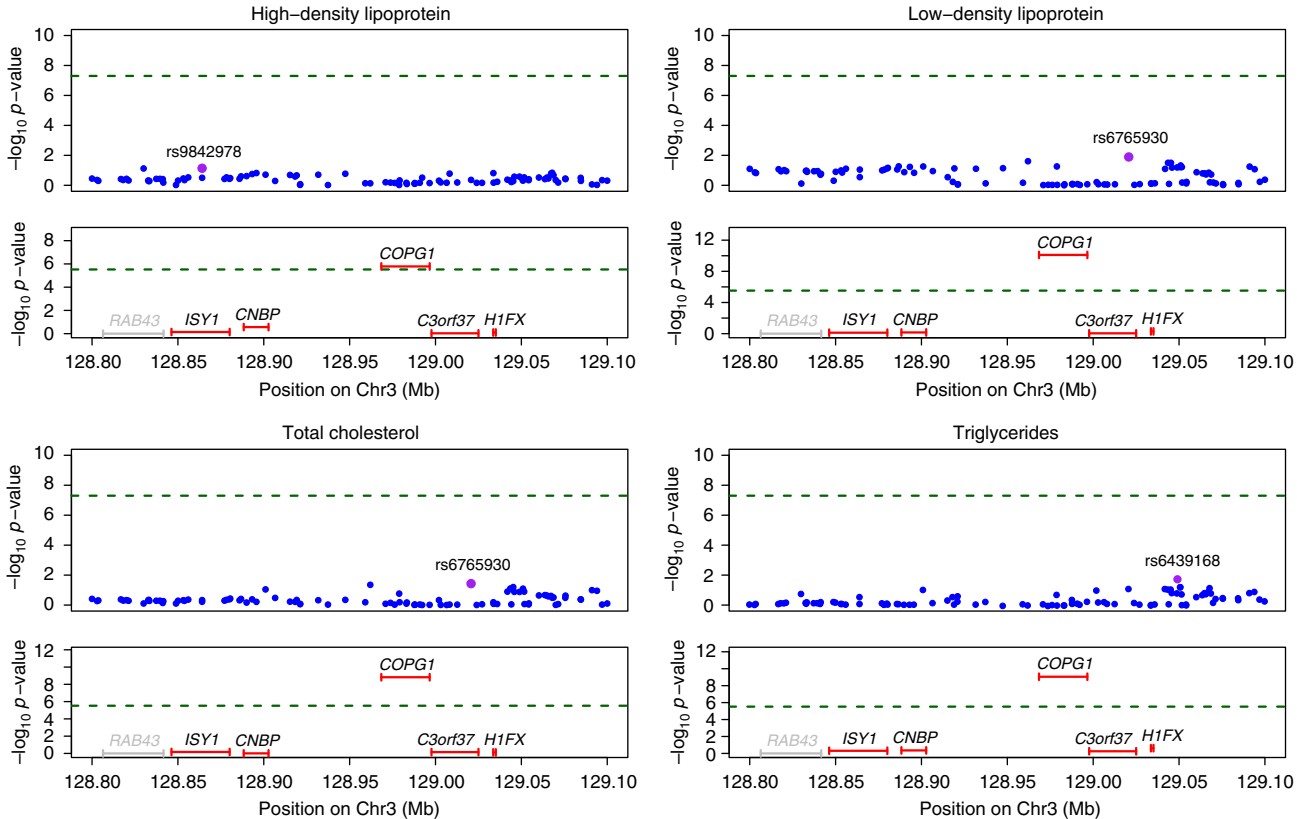

**Fig. 3** Significant association between *COPG1* and lipids traits. Regional association plot for GWAS and TWMR analyses for *COPG1* locus associated with high-density lipoprotein (HDL), low-density lipoprotein (LDL), total cholesterol (TC), and triglycerides (TG). Top panels show single-SNP association strengths ($y$ axis shows the $-\log_{10}(p$ value)) versus genomic position (on hg19/GRCh37 genomic build; $x$ axis) around the most significant SNP, indicated with a purple dot. Lower panels illustrate gene causal association strengths ($y$ axis shows the $-\log_{10}(p$ value)) versus genomic position. Genes highlighted in gray were not tested (i.e., they do not pass the heterogeneity test or not have significant eQTLs). Green dotted lines represent GWAS and TWMR significance thresholds

identified by LD score regression we found positive correlation between schizophrenia and ulcerative colitis ($\hat{\rho}_E = 0.08$, FDR $= 1.26 \times 10^{-03}$) in directional agreement with genetic correlation reported in a previous study[56] and supporting the molecular evidence for an autoimmune etiology for a fraction of schizophrenia cases. We also observed negative correlation between height and low-density lipoprotein cholesterol ($\hat{\rho}_E = -0.07$, FDR $= 6.71 \times 10^{-3}$), confirming the results of a previous study[57].

**Tissue-specific effects**. As many traits manifest themselves only in certain tissues, it is important to integrate data from the tissue of interest for the studied phenotype when trying to interpret GWAS results using gene expression as an intermediate phenotype. For this reason, we performed tissue-specific TWMR analyses using the eQTLs identified by GTEx (Genotype Tissue Expression Project)[29], which provides a unified view of genetic effects on gene expression across 48 human tissues. Despite the fact that sharing eQTLs (in consistent effect direction) across tissues is very common[58], there are many tissue-specific eQTLs.

For practical reasons we performed the tissue-specific analyses only for four phenotypes for which the key tissue is well known: CAD (artery), CD (intestine), LDL (liver), and T2D (pancreas). Among our results (Supplementary Data 2–5) we found *MRAS* and *PHACTR1*, associated with coronary artery disease (CAD), showing significant association in arterial tissues, which contribute most to the genetic causality of this trait[59] (*MRAS*: $P_{\text{TWMR}} = 2.04 \times 10^{-7}$ in tibial artery and $P_{\text{TWMR}} = 1.22 \times 10^{-6}$ in aorta,

*PHACTR1*: $P_{\text{TWMR}} = 4.80 \times 10^{-9}$ in tibial artery) (Fig. 5). Interestingly, no gene showed a significant causal effect in the other tissues, including whole blood using the large dataset from eQTLGen Consortium. *MRAS* and *PHACTR1* have significant eQTLs in other tissues but none of these effects was associated with the diseases, confirming that the disease-relevant eQTLs were tissue specific.

We also confirm *SORT1* being causal for LDL in line with previous findings[31]. Tissues that are not causal for LDL, like skin, pituitary, and testis[59], incorrectly pointed to *PSRC1* as the most likely putative causal gene and only liver pointed to *SORT1* as strong candidate gene (Supplementary Fig. 12 and Supplementary Data 4). This result confirms, once again, the importance to identify the relevant tissue(s) for the studied phenotype before looking for the causal gene.

## Discussion

We presented a powerful approach to perform TWMR analysis with multiple instruments and multiple exposures to identify genes with expression causally associated with complex traits. By increasing the statistical power through the integration of gene–expression and GWAS data, this method enables the prioritization of genes in known or novel associated regions and the identification of loci missed by conventional GWAS. We showed its efficacy through extensive analysis performed in 43 phenotypes. Whereas we have space to present only selected interesting findings, the readers will find the full results in the Supplementary Data 1–5. Exploiting UKBB data we showed that

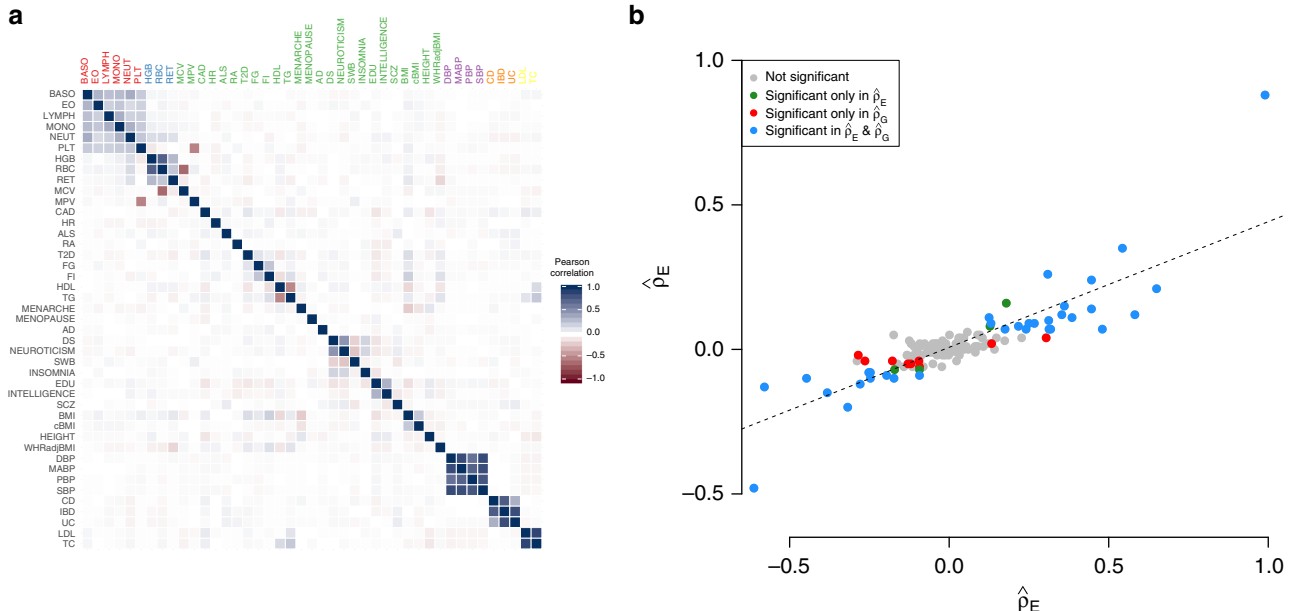

**Fig. 4** Genetic trait correlation at the level of gene expression. **a** Expression correlation from TWMR results among 43 traits. For the 2974 independent genes and each pair of traits we calculated the Pearson's correlation ($\hat{\rho}_E$) between the $Z$ scores (proportional to the standardized causal effects). Darker colors represent higher correlations, with blue and red corresponding to positive and negative associations, respectively. **b** Linear relationship between the expression correlation ($\hat{\rho}_E$) and the genetic correlation ($\hat{\rho}_G$) obtained from LD Score Regression. We selected the traits analyzed by our study and Bulik-Sullivan et al.[52] and for each pair of traits we compared the two correlations. Gray dots represent non-significant trait pairs, blue dots represent trait pairs significant for both correlation and red and green ones correspond to pairs of traits significant only in ($\hat{\rho}_G$) or ($\hat{\rho}_E$), respectively. The dotted line represents the regression line

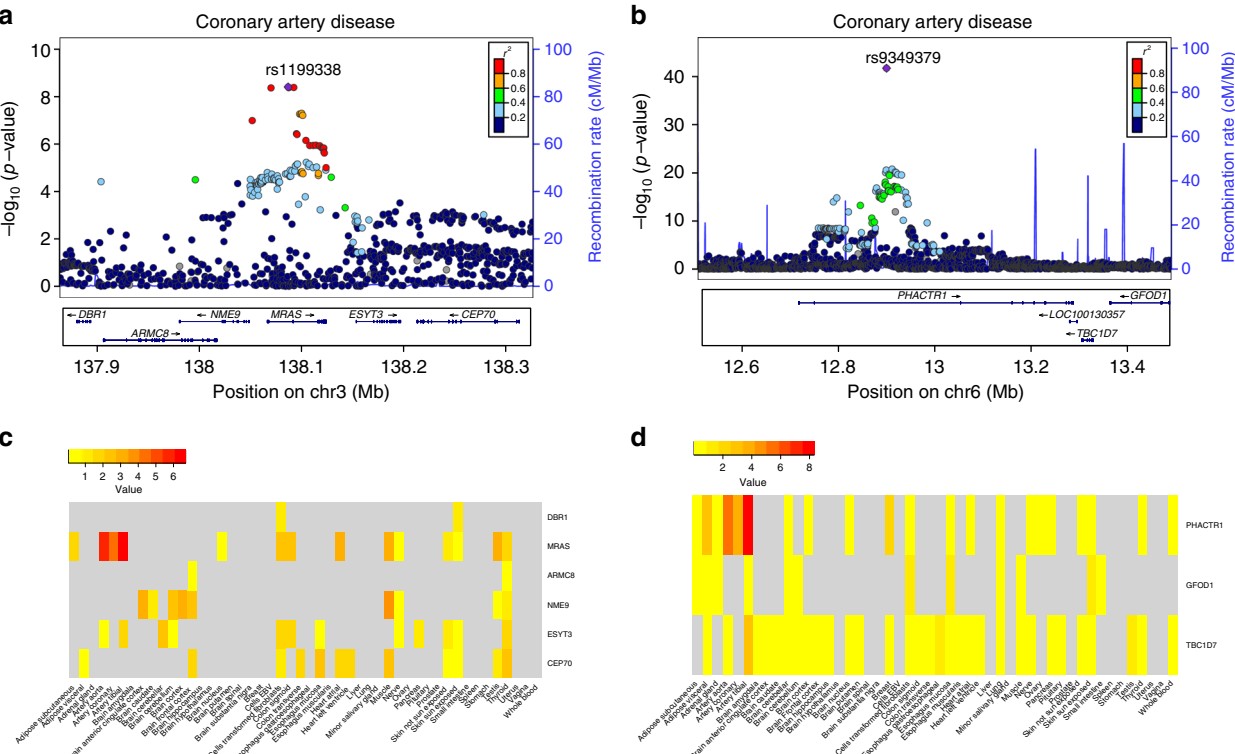

**Fig. 5** Tissue-specific effects of *MRAS* and *PHACTR1* on coronary artery disease. **a,b** Top. Association plot showing genome-wide significant loci for coronary artery disease in the *MRAS* **a** and *PHACTR1* **b** gene regions. Representation of single-SNP association strengths (*y* axis shows the $-\log_{10} p$ value) versus genomic positions (on hg19/GRCh37 genomic build; *x* axis) around the most significant SNP, indicated with a purple dot. Other SNPs in the region are color coded to reflect their LD with the top SNP (according to pair-wise $r^2$ values from 1000 Genomes Project phase 3 haplotypes). Bottom. Genes and position of exons and respective transcriptional strands. **c,d** Tissue-specific causal effects. Genes listed on *y* axis; tissues listed on *x* axis. Darker points correspond to stronger associations. Genes highlighted in gray were not tested. *MRAS* and *PHACTR1* show significant association only in arterial tissues

in most cases these new loci harbor true signals and will be eventually found by larger GWAS in the future.

Starting from our prioritized genes, we shed light on the shared genetic architecture of complex traits, and we estimated that about half of the genetic correlation is given by the gene–expression in whole blood, reinforcing the view that many risk variants affect complex traits through changes to gene regulation. Blood is, of course, not necessarily the causal tissue, more probably gene expression in blood may be an often good-enough proxy for the expression levels in more-relevant tissues[58].

Like all methods, our approach has its limitations, which need to be considered when interpreting results. The putative causal associations reported in this study are not definitive. They provide a prioritized list of candidate genes for future follow-up studies and also shed light on possible biological mechanisms of complex traits. This list of candidate genes will be more accurate in the future when much larger eQTL data sets will become available. Using larger eQTL data increases the number of instruments, which can either make the evidence for non-zero causal effect stronger or (under the null, in case the instruments with largest effect were pleiotropic) additional instruments can push the causal effect estimate toward zero. For example, we interrogate the 71 loci claimed significant for height in[25] and we observed that seven of them do not show any significant effect in our analyses ($P_{TWMR} > 0.05$). However, six out of these seven unconfirmed loci turn out to be significant when applying our method to the same small eQTL data set[26]. Of note, our approach revealed 263 new loci associated with height. Increased sample size will be crucial to detect SNPs associated with the gene–expression, whereas the possibility to interrogate more tissues could unravel causative genes, which expression is not well recapitulated in whole blood, the tissue in which the biggest human eQTL studies have been performed so far. Indeed, we failed to replicate the known association of *FTO* region with BMI because the effects of the *FTO* SNP on *IRX3* and *IRX5* are specific to primary adipocytes[49], a tissue not tested in our analyses. Furthermore, when conducting a tissue-specific analysis using GTEx data, we had considerably lower power to detect causal gene associations because for any given tissue, the sample size is more than 30 times lower than in the blood eQTL data set, ranging from 80 (brain-substantia nigra) to 491 (muscle) individuals. This represents an insurmountable limitation upon trying to identify tissue-specific causal genes given the limited number of eGenes shared between the tissues (Supplementary Fig. 13). In line with previous work[58] we observe that as long as a gene is expressed in blood, its eQTLs are fairly similar to its eQTLs in other tissues. We hypothesize that the tissue specificity of causal genes is mostly given whether the gene is expressed only in a given tissue. Moreover, our work provocatively suggests that large eQTL analyses such as GTEx with increased sample size and number of assessed tissues could pinpoint to causative genes more efficiently than increasing GWAS sampling.

Another shortcoming of our approach is that using current eQTL data only 16K eGenes are testable, which substantially decreases power to detect enrichment of our prioritized gene-set in relevant pathways and regulatory networks. We expect that larger eQTLs studies will allow identifying additional eGenes, resulting in stronger enrichment when using causally associated gene-sets rather than selecting genes based on physical proximity to GWAS hits. Still, many genes lead to disease not through change in their gene expression, but via modification of the RNA or protein sequence, mechanisms our approach is blind to.

Further limitations of our study are the violations of the MR assumptions. In particular, horizontal pleiotropy and indirect effects of the instruments on the exposures can substantially bias causal effect estimates. Although *cis*-eQTLs are thought to have direct effect of gene expression, we made particular effort to protect our results from potential biases, such as using multiple instruments, testing for effect heterogeneity and including other gene expression traits as exposures, through which potential pleiotropy may act. We think that accounting for pleiotropy, whenever the relevant trait is available, is a better approach than directly excluding violating instruments. In fact, with increasing GWAS study size, more SNPs will be excluded owing to evidence for mild pleiotropy, thereby reducing MR power.

Eventually, it is important to mention that, although in this study we used eQTLs data from gene–expression, our MR approach can be applied to other "omics" (e.g., methylation, metabolomics, proteomics) data. Indeed, our method only requires summary statistics from GWAS and any kind of exposure partnered with LD estimates, demonstrating once more the power of a carefully combined analysis of existing data to illuminate biological mechanism underlying complex traits and help the design of functional experiments.

## Methods

**TWMR analysis**. MR is a statistical method that uses genetic variants, so-called instrumental variables (IVs), to estimate the causal effect of an exposure on an outcome.

The IVs used in MR must verify three strong assumptions:

i.   They are associated with the exposure.
ii.  They are independent of any confounder of the exposure–outcome association.
iii. They are conditionally independent of the output given the exposure.

These assumptions imply that the genetic variants (IVs) have a causal effect on the outcome only via the risk factor. Although the first assumption can be easily verified, the second one is impossible to confirm as not all confounders are known, and the third requires instrument–exposure–outcome data measured in the same sample and is often violated by pleiotropy. Most probably, the third assumption is almost always violated in practice, but can be replaced by the weaker InSIDE assumption (instrument strength independent of the strength of the pleiotropy) when multiple independent instruments are available. Although we use multiple independent instruments, these may share mechanisms as they belong to the same cis region, which may lead to the violation of the InSIDE assumption[20]. In such regions, pleiotropic effects of the genetic variants on the outcome could act via a confounder owing to haplotype effects. One solution could be the inclusion of trans-eQTLs as instruments, but many of these are strong cis-eQTLs for other genes and hence much more likely to violate the second assumption of MR.

As many significant ($P_{eQTL} < 1.83 \times 10^{-5}$) eQTLs are in high LD with other nearby eQTLs, we pruned the eQTLs results using a stepwise selection procedure[60] on the basis of conditional $P_{eQTL}$ to select, for each gene, independent eQTLs. For each gene, using inverse–variance weighted method for summary statistics[34], we defined the causal effect of the gene expression on the outcome as

$$\widehat{\alpha} = (\mathbf{E'C}^{-1}\mathbf{E})^{-1}(\mathbf{E'C}^{-1}\mathbf{G}) \qquad (2)$$

Here **E** is a $n \times k$ matrix that contains the effect size of $n$ SNPs on $k$ gene expressions (these estimates come from an eQTL study). An individual SNP may affect a phenotype via different genes, therefore we estimated the causal effect jointly to allow for this correlation: starting from the IVs of the gene $i$, we included in the model all independent genes $e_2$, …., $e_k$ for which the IVs are significant eQTLs and all the independent significant eQTLs for those genes. **G** is a vector of length $n$ that contains the effect size of each SNP on the phenotype (these estimates come from the publicly available GWAS summary statistics). **C** is the pair-wise LD matrix between the $n$ SNPs. Here, the LD was based on UK10K reference panel[28]. To evaluate whether this panel approximates LD matrices derived from other European cohorts sufficiently well, we compared results when using LD estimates from 1000G-EUR[27] and we saw very good concordance ($r = 0.96$) between the causal effect estimates using different reference panels to estimate LD (Supplementary Fig. 15).

We ran a multivariable MR analysis for each ~ 16,000 gene, where we conditioned its causal effect on the potential causal effects of all of its neighboring genes. Let us consider now one focal gene. We need to select instrument SNPs and exposure genes for the multivariate MR analysis that is destined to elucidate the focal gene's multivariate causal effect on the outcome. To this end, we first consider all the independent eQTLs for the focal gene with conditional $P < 1 \times 10^{-3}$. Next, we include as exposures all the genes for which the selected SNPs are eQTLs. Finally, we extend the instruments to include all SNPs that are eQTLs for any of the exposure genes. Note that genes that do not share eQTLs with the focal gene do not alter the focal gene's multivariate causal effect, hence do not need to be considered here. To avoid numerical instability in our multiple regression model, we pruned

SNPs that are in high LD ($r^2 > 0.1$) (Fig. 1a). The variance of $\boldsymbol{\alpha}$ can be approximated by the Delta method[61].

$$var(\hat{\alpha}) = \left(\frac{\partial\hat{\alpha}}{\partial \boldsymbol{E}}\right)^2 * var(\boldsymbol{E}) + \left(\frac{\partial\hat{\alpha}}{\partial \boldsymbol{G}}\right)^2 * var(\boldsymbol{G}) + \left(\frac{\partial\hat{\alpha}}{\partial \boldsymbol{E}}\right) * \left(\frac{\partial\hat{\alpha}}{\partial \boldsymbol{G}}\right) * cov(\boldsymbol{E}, \boldsymbol{G}) \quad (3)$$

where $cov(\boldsymbol{E}, \boldsymbol{G})$ is 0 if $\boldsymbol{E}$ and $\boldsymbol{G}$ are estimated from independent samples (or if $\boldsymbol{E}$ and $\boldsymbol{G}$ are independent).

We defined the causal effect Z-statistic for gene $i$ as $\hat{\alpha}_i / SE(\hat{\alpha}_i)$, where $SE(\hat{\alpha}_i) = \sqrt{var(\hat{\alpha})_{i,i}}$.

**Applying TWMR to GWAS and eQTL summary results**. We applied TWMR to test each gene across the human genome for causal association between a phenotype and the expression level using summary statistics from GWAS and eQTLs studies.

We cover 43 traits by using publicly available GWAS summary statistics (i.e., for each SNP we extracted the estimated univariate effect size and its standard error) from the most recent meta-analyses. The traits analyzed in this study are listed in Supplementary Table 1.

All summary statistics were downloaded from the NIH Genome-wide Repository of Associations Between SNPs and Phenotypes (https://grasp.nhlbi.nih.gov/).

We used only SNPs on autosomal chromosomes and available in the UK10K reference panel, in order to be able to estimate the LD among these SNPs. We removed SNPs that were strand ambiguous, as well as those in the major histocompatibility complex region (chr6:26.2–33.8 Mb).

Cis-eQTL data were obtained from the eQTLGen Consortium (31,684 whole blood samples) and the GTEx[29], which includes 48 tissues collected from 11,688 post-mortem biopsies from 635 individuals (see Supplementary Table 2 for sample size per tissue).

**Simulation analyses**. To test if the multi-gene approach gives more accurate estimates than single-gene approach, we performed simulation analyses. We simulated a region containing 30 SNPs and three genes. We simulated 10,000 individuals for the eQTLs data set and another 100,000 for the GWAS data set. For each dataset, the genotypes of the SNPs were simulated from a binomial distribution $s \sim B(2, q)$, $q \sim U(0.05, 0.5)$. The SNPs genotypes ($s$) were then standardized as $z = \frac{s-2q}{\sqrt{2q(1-q)}}$.

For each SNP, we simulated its degree of pleiotropy ($l$) from a Poisson distribution ($\lambda = 0.4$) (i.e., $\lambda$ is the mean of the number of genes affected by the SNP). The effects of the SNPs ($\beta_E$) on the gene expression were estimated from a normal distribution ($\beta_E|\beta_E \neq 0) \sim N(0, \sigma^2)$, where $\sigma^2 = h_j^2/l$, $h_j^2$ is the heritability of the gene $j$ estimated from a uniform distribution $h_j^2 \sim U(0.01, 0.4)$.

We simulated the expression of gene $j$ based on the model

$$E_j = \sum_{i=1}^{30} \beta_{ij} z_{\cdot i} + \gamma_E c + \varepsilon_j \quad (4)$$

where $\gamma_E = 0.2$, $c \sim N(0,1)$, $\varepsilon_j \sim N(0, \sigma^2)$ and $\sigma^2 = 1 - \sum_{i=1}^{30} \beta_{ij}^2 - \gamma_E^2$.

We simulated the phenotype based on the model

$$T = \sum_{j=1}^{3} E_j \alpha_j + \gamma_T c' + \varepsilon \quad (5)$$

where $\alpha \sim N(0, 0.05)$, $\gamma_T = 0.1$, $c' \sim N(0,1)$, $\varepsilon_j \sim N(0, \sigma^2)$ and $\sigma^2 = 1 - \sum_{i=1}^{3} \alpha_i^2 - \gamma_T^2$.

We repeated the simulation 1000 times. In each simulation replicated, we estimated $\hat{\beta}_E$ (the effect of the SNPs on the gene expression) and $\hat{\beta}_G$ (the effect of the SNPs on the phenotype) by a linear regression analyses and performed the multi- and single- gene MR approach applying the formula described above. We also experimented with different values of pleiotropy (in Poisson distribution $\lambda = 0.2, 0.4, 0.6,$ and $0.8$).

To investigate the statistical power of TWMR, we generated $T$ under the assumption that $E_1$ has a causal effect on $T$ using the same method as above. We simulated three values of $\alpha_1$ (0.05, 0.03, and 0.01) and for each scenario we ran the simulations 1000 times. Similarly, we calculated the type I error for the two approaches under the assumption of no causal effect of $E_1$ on $T$ (i.e., $\alpha_1 = 0$) (Supplementary Figs. 1–6).

**Heterogeneity test**. The validity of the MR approach relies on three assumptions, of which the third assumption (no pleiotropy) is crucial as MR causal estimates will be biased if the genetic variants (IVs) have pleiotropic effects[18]. Our method assumes that all genetic variants used as instrumental variables affect the outcome only through gene expression and not through independent biological pathways.

In order to test for the presence of pleiotropy, we used Cochran's Q test[20,36].

In brief, we tested whether there is a significant difference between the MR-effect of an instrument (i.e., $\sum_{k=1}^{K} \alpha_k E_{i,k}$) and the estimated effect of that

instrument on the phenotype ($G_i$). We defined

$$d_i = G_i - \sum_{k=1}^{K} \alpha_k E_{i,k} \quad (6)$$

and its variance as

$$var(d_i) = var(G_i) + (E_i)^2 * var(\alpha) + var(E_i) * (\alpha)^2 + var(E_i) * var(\alpha) \quad (7)$$

We can test the deviation of each SNP using the following test statistic

$$T_i = \frac{d_i^2}{var(d_i)} \sim \chi_m^2 \quad (8)$$

In case of $P < 1 \times 10^{-4}$, we removed the SNP with largest $|d_i|$ and then repeated the test.

Performing 10 iterations of the heterogeneity test for height, we observed that after three iterations we corrected >90% of the genes and after 5 we reach a plateau. Thus, for practical reasons we decided to perform maximum three iterations for each phenotype (Supplementary Fig. 14) and loci still showing heterogeneity were discarded from further analysis.

**Correlation between genes**. A common observation in the analysis of gene expression is that often physically close genes show similar expression patterns[62]. To avoid numerical instability caused by near-colinearity in our multiple regression model and making choices between co-regulated genes, we removed one gene from each pair of genes with $r^2 \geq 0.4$. The correlation $r^2$ was estimated as Pearson's correlation between the Z scores of the shared, independent eQTLs.

This strategy is blind to causal pathways where Gene1 → Gene2 → trait. In such scenario, Gene1-eQTLs are also eQTLs for Gene2 and since their effect sizes are proportional (Gene1 → Gene2), they are highly correlated. It means that, as we are excluding highly correlated genes, Gene2 would be excluded (but its causal effect is estimated separately when Gene1 is excluded) when we estimate the causal effect of Gene1 on the trait and vice versa.

**Genetic trait correlation at the level of gene expression**. To estimate the phenotypic correlation between each pair of traits we calculated the Pearson's correlation between the Z scores across the set of 2,974 independent genes extracted from the 11,510 genes analyzed for all the 43 traits included in our analyses.

**Genome-wide association study in UKBB**. We ran genome-wide association study of BMI in 379,530 unrelated British individuals from UKBB using linear model association testing implemented in bgenie software[63].

We analyzed only SNPs included in UK10K reference panel and with a rs name. In total we analyzed 15,599,830 SNPs.

BMI was adjusted for sex, age, age$^2$, and 40 principal components.

**URLs**. For PLINK 1.90, see https://www.cog-genomics.org/plink2; for GCTA, see http://cnsgenomics.com/software/gcta/#Download; for GTEx Portal, see http://www.gtexportal.org/; for OMIM, see https://www.omim.org; for UK Biobank, see http://www.ukbiobank.ac.uk/; for BGENIE, see https://jmarchini.org/bgenie; for LOCUSZOOM, see http://locuszoom.org; for NHGRI-EBI GWAS Catalog, see http://www.ebi.ac.uk/gwas.

**Reporting summary**. Further information on research design is available in the Nature Research Reporting Summary linked to this article.

## Data availability

All summary statistics were downloaded from the NIH Genome-wide Repository of Associations Between SNPs and Phenotypes (https://grasp.nhlbi.nih.gov/). The software tools are available at the URLs above.

## Code availability

R-code for performing TWMR analyses is available at https://github.com/eleporcu/TWMR.

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

## Acknowledgements

This work was supported by grants from the Swiss National Science Foundation (31003A_143914 and 32003B_173092 to ZK and 31003A_160203 to AR) and the

Horizon2020 Twinning project ePerMed (692145 to AR). This research has been conducted using the UK Biobank Resource (#16389). The funders had no role in study design, data collection and analysis, decision to publish, or preparation of the manuscript.

## Author contributions

Z.K. conceived and designed the study. Z.K., A.R. and E.P. derived the theories. E.P. performed statistical analyses. Z.K., K.L. and E.P. designed the method. S.R., K.L. and F.A.S. contributed by providing statistical support. All the authors contributed by providing advice on interpretation of results. E.P, A.R. and Z.K. wrote the manuscript with the participation of all authors. eQTLGen Consortium and BIOS Consortium provided eQTLs data. Members of the eQTLGen Consortium and BIOS Consortium are ordered alphabetically.

## Additional information

**Competing interests:** The authors declare no competing interests.

## eQTLGen Consortium

Mawussé Agbessi[6], Habibul Ahsan[7], Isabel Alves[6], Anand Andiappan[8], Wibowo Arindrarto[9], Philip Awadalla[6], Alexis Battle[10,11], Frank Beutner[12], Marc Jan Bonder[13,14,15], Dorret Boomsma[16], Mark Christiansen[17], Annique Claringbould[13], Patrick Deelen[13,18], Tõnu Esko[19], Marie-Julie Favé[11], Lude Franke[13], Timothy Frayling[20], Sina A. Gharib[17,21], Gregory Gibson[22], Bastiaan T. Heijmans[9], Gibran Hemani[23], Rick Jansen[24], Mika Kähönen[25], Anette Kalnapenkis[19], Silva Kasela[19], Johannes Kettunen[26], Yungil Kim[11,27], Holger Kirsten[28], Peter Kovacs[29], Knut Krohn[30], Jaanika Kronberg-Guzman[19], Viktorija Kukushkina[19], Bernett Lee[8], Terho Lehtimäki[31], Markus Loeffler[28], Urko M. Marigorta[22], Hailang Mei[9], Lili Milani[19], Grant W. Montgomery[32], Martina Müller-Nurasyid[33,34,35], Matthias Nauck[36], Michel Nivard[16], Brenda Penninx[25], Markus Perola[37], Natalia Pervjakova[19], Brandon L. Pierce[7], Joseph Powell[38], Holger Prokisch[39,40], Bruce M. Psaty[17,41,42], Olli T. Raitakari[43], Samuli Ripatti[44], Olaf Rotzschke[8], Ashis Saha[11], Markus Scholz[28], Katharina Schramm[35,36], Ilkka Seppälä[31], Eline P. Slagboom[9], Coen D.A. Stehouwer[45], Michael Stumvoll[46], Patrick Sullivan[47], Peter A.C. 't Hoen[48], Alexander Teumer[49], Joachim Thiery[50], Lin Tong[7], Anke Tönjes[46], Jenny van Dongen[16], Maarten van Iterson[9], Joyce van Meurs[51], Jan H. Veldink[52], Joost Verlouw[51], Peter M. Visscher[32], Uwe Völker[53], Urmo Võsa[13,19], Harm-Jan Westra[13], Cisca Wijmenga[13], Hanieh Yaghootkar[20], Jian Yang[32,54], Biao Zeng[22] & Futao Zhang[32]

[6]Computational Biology, Ontario Institute for Cancer Research, Toronto, ON, Canada. [7]Department of Public Health Sciences, University of Chicago, Chicago, IL, USA. [8]Singapore Immunology Network, Agency for Science, Technology and Research, Singapore, Singapore. [9]Department of Biomedical Data Sciences, Leiden University Medical Center, Leiden, The Netherlands. [10]Department of Biomedical Engineering, Johns Hopkins University, Baltimore, MD, USA. [11]Department of Computer Science, Johns Hopkins University, Baltimore, MD, USA. [12]Heart Center Leipzig, Universität Leipzig, Leipzig, Germany. [13]Department of Genetics, University Medical Centre Groningen, Groningen, The Netherlands. [14]European Molecular Biology Laboratory, European Bioinformatics Institute, Wellcome Genome Campus, Hinxton, UK. [15]Genome Biology Unit, European Molecular Biology Laboratory, Heidelberg, Germany. [16]Department of Biological Psychology, Vrije Universiteit Amsterdam, Amsterdam, The Netherlands. [17]Cardiovascular Health Research Unit, University of Washington, Seattle, WA, USA. [18]Genomics Coordination Center, University Medical Centre Groningen, Groningen, The Netherlands. [19]Estonian Genome Center, Institute of Genomics, University of Tartu, Tartu 51010, Estonia. [20]Exeter Medical School, University of Exeter, Exeter, EX2 5DW, UK. [21]Department of Medicine, University of Washington, Seattle, WA, USA. [22]School of Biological Sciences, Georgia Tech, Atlanta, GA, USA. [23]MRC Integrative Epidemiology Unit, University of Bristol, Bristol, UK. [24]Department of Psychiatry and Amsterdam Neuroscience, Amsterdam UMC, Vrije Universiteit, Amsterdam, The Netherlands. [25]Department of Clinical Physiology, Tampere University Hospital and Faculty of Medicine and Health Technology, Tampere University, Tampere, Finland. [26]Centre for Life Course Health Research, University of Oulu, Oulu, Finland. [27]Genetics and Genomic Science Department, Icahn School of Medicine at Mount Sinai, New York, NY, USA. [28]Institut für Medizinische Informatik, Statistik und Epidemiologie, LIFE – Leipzig Research Center for Civilization

Diseases, Universität Leipzig, Leipzig, Germany. [29]IFB Adiposity Diseases, Universität Leipzig, Leipzig, Germany. [30]Interdisciplinary Center for Clinical Research, Faculty of Medicine, Universität Leipzig, Leipzig, Germany. [31]Department of Clinical Chemistry, Fimlab Laboratories and Finnish Cardiovascular Research Center-Tampere, Faculty of Medicine and Health Technology, Tampere University, Tampere, Finland. [32]Institute for Molecular Bioscience, University of Queensland, Brisbane, QLD, Australia. [33]Institute of Genetic Epidemiology, Helmholtz Zentrum München - German Research Center for Environmental Health, Neuherberg, Germany. [34]Department of Medicine I, University Hospital Munich, Ludwig Maximilian's University, München, Germany. [35]DZHK (German Centre for Cardiovascular Research), partner site Munich Heart Alliance, Munich, Germany. [36]Institute of Clinical Chemistry and Laboratory Medicine, University Medicine Greifswald, Greifswald, Germany. [37]National Institute for Health and Welfare, University of Helsinki, Helsinki, Finland. [38]Garvan Institute of Medical Research, Garvan-Weizmann Centre for Cellular Genomics, Sydney, Australia. [39]Institute of Human Genetics, Helmholtz Zentrum München, Neuherberg, Germany. [40]Institute of Human Genetics, Technical University Munich, Munich, Germany. [41]Departments of Epidemiology, Medicine, and Health Services, University of Washington, Seattle, WA, USA. [42]Kaiser Permanente Washington Health Research Institute, Seattle, WA, USA. [43]Centre for Population Health Research, Department of Clinical Physiology and Nuclear Medicine, Turku University Hospital and University of Turku, Turku, Finland. [44]Statistical and Translational Genetics, University of Helsinki, Helsinki, Finland. [45]Department of Internal Medicine, Maastricht University Medical Centre, Maastricht, The Netherlands. [46]Department of Medicine, Universität Leipzig, Leipzig, Germany. [47]Department of Medical Epidemiology and Biostatistics, Karolinska Institutet, Stockholm, Sweden. [48]Center for Molecular and Biomolecular Informatics, Radboud Institute for Molecular Life Sciences, Radboud University Medical Center Nijmegen, Nijmegen, The Netherlands. [49]Institute for Community Medicine, University Medicine Greifswald, Greifswald, Germany. [50]Institute for Laboratory Medicine, LIFE – Leipzig Research Center for Civilization Diseases, Universität Leipzig, Leipzig, Germany. [51]Department of Internal Medicine, Erasmus Medical Centre, Rotterdam, The Netherlands. [52]Department of Neurology, University Medical Center Utrecht, Utrecht, The Netherlands. [53]Interfaculty Institute for Genetics and Functional Genomics, University Medicine Greifswald, Greifswald, Germany. [54]Institute for Advanced Research, Wenzhou Medical University, Wenzhou, Zhejiang 325027, China

## BIOS Consortium

Wibowo Arindrarto[55], Marian Beekman[56], Dorret I. Boomsma[16], Jan Bot[57], Joris Deelen[56], Patrick Deelen[13], Lude Franke[13], Bastiaan T. Heijmans[56], Peter A.C. 't Hoen[58], Bert A. Hofman[59], Jouke J. Hottenga[16], Aaron Isaacs[60], Marc Jan Bonder[13], P. Mila Jhamai[3], Rick Jansen[24], Szymon M. Kielbasa[61], Nico Lakenberg[56], René Luijk[56], Hailiang Mei[62], Matthijs Moed[56], Irene Nooren[57], René Pool[16], Casper G. Schalkwijk[63], P. Eline Slagboom[56], Coen D.A. Stehouwer[45], H. Eka D. Suchiman[56], Morris A. Swertz[18], Ettje F. Tigchelaar[13], André G. Uitterlinden[51], Leonard H. van den Berg[57], Ruud van der Breggen[56], Carla J.H. van der Kallen[63], Freerk van Dijk[18], Jenny van Dongen[16], Cornelia M. van Duijn[60], Michiel van Galen[58], Marleen M.J. van Greevenbroek[63], Diana van Heemst[64], Maarten van Iterson[56], Joyce van Meurs[51], Jeroen van Rooij[51], Peter van't Hof[55], Erik.W. van Zwet[61], Martijn Vermaat[58], Jan H. Veldink[52], Michael Verbiest[51], Marijn Verkerk[51], Cisca Wijmenga[13], Dasha V. Zhernakova[13] & Sasha Zhernakova[13]

[55]Sequence Analysis Support Core, Leiden University Medical Center, Leiden, The Netherlands. [56]Molecular Epidemiology Section, Department of Medical Statistics and Bioinformatics, Leiden University Medical Center, Leiden 2333 ZA, The Netherlands. [57]SURFsara, Amsterdam, The Netherlands. [58]Department of Human Genetics, Leiden University Medical Center, Leiden, The Netherlands. [59]Department of Epidemiology, ErasmusMC, Rotterdam, The Netherlands. [60]Department of Genetic Epidemiology, ErasmusMC, Rotterdam, The Netherlands. [61]Medical Statistics Section, Department of Medical Statistics and Bioinformatics, Leiden University Medical Center, Leiden, The Netherlands. [62]Sequence Analysis Support Core, Leiden University Medical Center, Leiden, The Netherlands. [63]Department of Internal Medicine and School for Cardiovascular Diseases (CARIM), Maastricht University Medical Center, Maastricht, The Netherlands. [64]Department of Gerontology and Geriatrics, Leiden University Medical Center, Leiden, The Netherlands

