## [Peer Review File · Nature Communications]

Editorial Note: This manuscript has been previously reviewed at another journal that is not operating a transparent peer review scheme. This document only contains reviewer comments and rebuttal letters for versions considered at Nature Communications. Mentions of the other journal have been redacted.

Reviewers' Comments:

Reviewer #1:

Remarks to the Author:

Thank you to the authors for addressing my points. I do like this paper. But my apologies for dragging this out but still a few things that I think should be clarified. I know that some of the below will look a bit pedantic, but I very regularly speak to people who apply these types of methods without full understanding of the implications, and with this paper which I hope will be quite impactful you have a great opportunity to really lay things out correctly. I hope that's reasonable.

In (3.) I had packaged up several things and not all was addressed, but I think it's important. To rephrase - the point of MR is to avoid having to know unmeasured confounding because instruments are used instead. But the multivariable approach seeks to address unintended causal relationships through inclusion of traits through which pleiotropy passes - this is akin to the very problems that MR is trying to avoid because if you have unmeasured pleiotropy pathways then you can't account for them. In the case of

SNP -> U

GENE <- U -> Trait

the use of SNP as instrument for GENE will lead to GENE being determined as causal for Trait unless U is present in the analysis (and technically there are at least two instruments, 1+ for U and 1+ for GENE or 2+ for U). Multivariable MR used with a view to avoid this situation has exactly the same conceptual limitations as observational associations.

I don't think (7.) has been addressed. In addition, in terms of the modified text, the authors state 'Zhu et al [11] developed a co-localization method in a summary-based MR analyses framework (SMR) to test whether the effects of genetic variants on a phenotype are mediated by gene expression.' genetic colocalisation in MR development is not as described - the point is that genetic colocalisation is all you can do when you have a single locus (i.e. for omic variables). The use of genetic colocalisation for MR of gene expression came about AFTER multi-locus MR had already been in wide use, because it was known that it couldn't avoid horizontal pleiotropy in the single locus case, but it could try to avoid the instance where there were separate causal variants for the gene and the trait.

GSMR of course came later (which is basically MR-IVW) and it is what you do when you have multiple independent variants (across the genome) - and it's application was to trait-trait associations not gene-trait. If you apply GSMR / IVW to gene-trait analysis in order to boost the causal interpretability you have to make a strong assumption that all conditionally independent variants in the same locus are akin to meta analysing independent randomised control trials. I think this is the claim that is being made and should be stated as such.

In (6.) I think you need to further clarify this point. The genetic confounding of X-Y is not talked about so much in the literature because MR is typically applied to complex traits that have very many associations. It would be rare to find a situation where all the instruments for complex trait X arise through some single genetic confounder. But mostly there will be a variety of confounders, and this gives rise to heterogeneity which are attended to by usual pleiotropy robust approaches. For the case of this analysis, genetic confounding of X-Y is of high concern because you only have one instrumenting locus for a gene. So, its importance is elevated in the context of instrumenting gene expression.

In (9.) this figure is helpful. I think you clarify in step 2 they are conditionally independent variants. Are you using the adjusted or raw effect estimates to feed into the MR model (i.e. does the conditioning of SNP-exposure effect sizes happen in the MR estimation stage)?

(10.) Thank you for looking into this issue - is it the case that the example given for rs11861657 is for one multivariable MR analysis, in which there were 24 genes modelled against an outcome? This raises a couple more questions:

1. How many conditionally independent eQTLs are there for each gene?
2. Across all multivariable MR analyses, how many genes are typically included?

I think it's important to include (14.) in the paper. I am sure the authors don't see the method as a panacea for mapping causal genes, and I am sure neither the editors nor any readers should either. It is an important step forward, but understanding the limitations and where things might be giving confusing results etc is crucial for making appropriate interpretations.

Reviewer #2:

Remarks to the Author:

This may be my lack of understanding rather than the authors' presentation, but I am still struggling to understand what is going on in this manuscript. It's not helped by the structure imposed by the journal of putting the Methods section at the end of the manuscript. But even in this structure, it would help enormously if the authors could signpost the manuscript better. And not only the readability of the manuscript, but for me also the clarity of the science. Currently the introduction says: "We have a tool. We are going to hit data with the tool." Which is fair enough as far as it goes, but I have no idea what the authors expect to show by applying the tool to the data. Are there examples - positive controls or negative controls - which the authors could apply the method to and show that the method is working as expected? There is some hint of this in the manuscript, but it's all presented post hoc - for example the "abnormal skeletal growth syndrome" example on page 6. It would help the science as well as the presentation if this were set up as - here is our hypothesis, here is the data we used to test the hypothesis, and here is the result of the hypothesis test. Currently, it reads like the authors tried some things, and these are their observations - like a lab notebook rather than a hypothesis-led piece of science. There's other aspects of the analysis that again are described on the fly rather than in a principled way. The simulation - what specific properties of the method are the authors trying to show in the simulation study? There's currently no motivation for this. The section on "trait correlation" - there's no mention of this in the introduction or the methods. It may be a worthwhile thing to do, but it gives the impression that the authors did this because they could, rather than having a prior motivation for doing this. Similarly for the section on "tissue-specific effects" - where did that come from? Similarly for the part on discovering new loci - I thought the point of the method was to link gene expression to phenotypes? There even appears to be a new analysis introduced in the discussion: "For example, we interrogate the 71 loci...".

In short, it would really help if the introduction had a paragraph saying: "In this paper, we will do X, Y and Z" and described the motivation for doing X, Y and Z, the data to be used, the hypothesis they are testing, what they expect to see, and so on. Currently, I find it hard to follow what analyses have been performed, I don't get why the different analyses are performed, what they are showing or how I'm supposed to know if they were successful or not.

Figure 1 is really helpful for understanding how the method was performed, and how the authors selected which genes to include in the analysis. How were the authors certain that they included all

relevant genes in each analysis? Also, there appears to be some confusion about the pruning threshold ($LD < 0.1$ here, $r^2 < 0.4$ in the methods section).

"One way to guard against the violation of the third assumption is to use as many IVs possible, as the pleiotropic effect of each marker will cancel each other out under the INSIDE assumption (instrument strength independent of the strength of the pleiotropy)."

This is completely untrue. There's no reason why pleiotropic effects would cancel out, even under the InSIDE assumption (pleiotropic effects can be independent and not average to zero). And this is not how assumptions work - you cannot invoke an assumption to make a point: it's like saying that A is true because I have assumed that A is true. The argument is circular.

Reviewer #1 (Remarks to the Author):

Thank you to the authors for addressing my points. I do like this paper. But my apologies for dragging this out but still a few things that I think should be clarified. I know that some of the below will look a bit pedantic, but I very regularly speak to people who apply these types of methods without full understanding of the implications, and with this paper which I hope will be quite impactful you have a great opportunity to really lay things out correctly. I hope that's reasonable.

We thank the reviewer for his/her praise. We will do our best to clarify the parts that are still unclear.

In (3.) I had packaged up several things and not all was addressed, but I think it's important. To rephrase - the point of MR is to avoid having to know unmeasured confounding because instruments are used instead. But the multivariable approach seeks to address unintended causal relationships through inclusion of traits through which pleiotropy passes - this is akin to the very problems that MR is trying to avoid because if you have unmeasured pleiotropy pathways then you can't account for them. In the case of

*SNP \rightarrow U
GENE \leftarrow U \rightarrow Trait*

the use of SNP as instrument for GENE will lead to GENE being determined as causal for Trait unless U is present in the analysis (and technically there are at least two instruments, 1+ for U and 1+ for GENE or 2+ for U). Multivariable MR used with a view to avoid this situation has exactly the same conceptual limitations as observational associations.

We have not explained clearly our reasoning: observational associations suffer from confounding, which (univariate) MR can avoid under its assumptions. Indeed, instruments associated with the confounder (U) of the GENE-Trait association will suffer from the same problem as observational associations. One way to avoid this problem is to perform multivariable MR, which includes additional exposures in the analysis, hoping that some of them capture U. The key difference between this limitation of multivariable MR and observational associations is that for the former we would like to capture (account for) ONLY THOSE confounders that share genetic instruments with GENE, while any other additional confounder does not need to be measured/included in the analysis and it still yields unbiased causal effect estimate. Thus, we argue that multivariable MR has potentially less limitation than both univariable MR and (multivariate) observational association.

I don't think (7.) has been addressed. In addition, in terms of the modified text, the authors state 'Zhu et al [11] developed a co-localization method in a summary-based MR analyses framework (SMR) to test whether the effects of genetic variants on a phenotype are mediated by gene expression.' genetic colocalisation in MR development is not as

described - the point is that genetic colocalisation is all you can do when you have a single locus (i.e. for omic variables). The use of genetic colocalisation for MR of gene expression came about AFTER multi-locus MR had already been in wide use, because it was known that it couldn't avoid horizontal pleiotropy in the single locus case, but it could try to avoid the instance where there were separate causal variants for the gene and the trait.

GSMR of course came later (which is basically MR-IVW) and it is what you do when you have multiple independent variants (across the genome) - and its application was to trait-trait associations not gene-trait. If you apply GSMR / IVW to gene-trait analysis in order to boost the causal interpretability you have to make a strong assumption that all conditionally independent variants in the same locus are akin to meta analysing independent randomised control trials. I think this is the claim that is being made and should be stated as such.

We use a slightly different definition of colocalisation method than the reviewer. Typical colocalisation methods, such as COLOC [<https://journals.plos.org/plosgenetics/article?id=10.1371/journal.pgen.1004383>] are symmetrical for the two examined traits and do not attempt to seek evidence for causality. COLOC tests 5 models, none of which corresponds to the causal scenario: if trait1 → trait2 holds, SNPs could be in the same locus that are associated with both traits and some other SNPs that are associated only with trait2. None of the 5 models capture this. In our view, colocalisation methods (exemplified by COLOC) are not causal inference (nor MR) methods. Therefore, we would rather not talk about colocalisation methods at all, but distinguish single- and multi-locus MR methods. We have modified the text accordingly:

A conventional MR analysis estimates the causal effect of a risk factor (exposure) on an outcome by using genetic variant(s) that are (directly) associated only with the risk factor as instrumental variables [11]. Since mostly SNPs have small effects on phenotypes, increasing the number of instruments increases the statistical power [12]. If SNPs, exposure and outcome are all measured in the same sample, the causal effect of the risk factor in the outcome can be estimated using a 2-stage least squares [13] approach. However, large cohorts of this kind are rare rendering such approach heavily underpowered. It is often the case that the exposure and the outcome are available in different datasets, a situation for which two-sample MR methods have been developed [14]. The key advantage of using two-sample MR is that it only requires publicly available GWAS summary statistics [15]. In such an approach, only independent SNPs are considered and a fixed effects inverse variance weighted (IVW) meta-analysis is used to estimate the causal effect [16][17]. Since the IVW estimate is a weighted average of the effects from each SNP, if any of the SNPs shows horizontal pleiotropy then the causal effect estimate is biased. However, such pleiotropy introduces heterogeneity, which can be detected and SNPs contributing the most to the heterogeneity can be excluded [18][19][20], a good solution if the majority of the instruments are valid. Conversely, when some of the genetic variants in the analysis are not valid instruments, other MR approaches (i.e. MR-Egger [20], weighted median [21] or mode-based MR [22]) should be applied.

Although such methods provide a more robust estimate of the causal effect, they have less power to detect causal association.

Pleiotropy could alternatively be tackled using multivariable MR. If a variant exhibits horizontal pleiotropy, but we know its association(s) to some mediators of its indirect effect to the outcome, those mediators could be included as additional exposures and one can perform a multivariable MR, which can mitigate bias by jointly estimating the causal effects of all exposures on the outcome [23][24].

In (6.) I think you need to further clarify this point. The genetic confounding of X-Y is not talked about so much in the literature because MR is typically applied to complex traits that have very many associations. It would be rare to find a situation where all the instruments for complex trait X arise through some single genetic confounder. But mostly there will be a variety of confounders, and this gives rise to heterogeneity which are attended to by usual pleiotropy robust approaches. For the case of this analysis, genetic confounding of X-Y is of high concern because you only have one instrumenting locus for a gene. So, its importance is elevated in the context of instrumenting gene expression.

We agree with the reviewer that in our analyses the genetic confounding of X-Y can be an issue since we are performing single-locus analyses. We already pointed to this in the Methods section:

While we use multiple independent instruments, these may share mechanisms as they belong to the same cis region, which may lead to the violation of the INSIDE assumption [20]. In such regions, pleiotropic effects of the genetic variants on the outcome could act via a confounder due to haplotype effects. One solution could be the inclusion of trans-eQTLs as instruments, but many of these are strong cis-eQTLs for other genes and hence much more likely to violate the second assumption of MR.

In (9.) this figure is helpful. I think you clarify in step 2 they are conditionally independent variants. Are you using the adjusted or raw effect estimates to feed into the MR model (i.e. does the conditioning of SNP-exposure effect sizes happen in the MR estimation stage)?

We use GCTA only to select the independent SNPs and then we use the raw (univariate) effect estimates for such SNPs in the MR model – as the formula requires.

(10.) Thank you for looking into this issue - is it the case that the example given for rs11861657 is for one multivariable MR analysis, in which there were 24 genes modelled against an outcome?

Correct. It is an extreme example to show the high variability of the effect size of cis-eQTLs.

This raises a couple more questions:

1. How many conditionally independent eQTLs are there for each gene?
2. Across all multivariable MR analyses, how many genes are typically included?

The number of genes and SNPs included in each test is reported in Supplementary Table2. On average, for example for height, we included 8 SNPs and 2 genes in each MR model.

I think it's important to include (14.) in the paper. I am sure the authors don't see the method as a panacea for mapping causal genes, and I am sure neither the editors nor any readers should either. It is an important step forward, but understanding the limitations and where things might be giving confusing results etc is crucial for making appropriate interpretations.

Correct, we don't think our method is a panacea and as any MR method can lead to false positive and negative findings. Beyond the violation of the methodological assumptions, false negative associations can be due to testing an irrelevant tissue, as is the case for the effect of the *FTO* region on BMI. As requested by the reviewer, we added a sentence in the Discussion:

Indeed, we failed to replicate the known association of *FTO* region with BMI because the effects of the *FTO* SNP on *IRX3* and *IRX5* are specific to primary adipocytes [49], a tissue not tested in our analyses.

Reviewer #2 (Remarks to the Author):

This may be my lack of understanding rather than the authors' presentation, but I am still struggling to understand what is going on in this manuscript. It's not helped by the structure imposed by the journal of putting the Methods section at the end of the manuscript. But even in this structure, it would help enormously if the authors could signpost the manuscript better. And not only the readability of the manuscript, but for me also the clarity of the science. Currently the introduction says: "We have a tool. We are going to hit data with the tool." Which is fair enough as far as it goes, but I have no idea what the authors expect to show by applying the tool to the data. Are there examples - positive controls or negative controls - which the authors could apply the method to and show that the method is working as expected? There is some hint of this in the manuscript, but it's all presented post hoc - for example the "abnormal skeletal growth syndrome" example on page 6. It would help the science as well as the presentation if this were set up as - here is our hypothesis, here is the data we used to test the hypothesis, and here is the result of the hypothesis test.

We agree with the reviewer that the logic of the manuscript may not have been clearly presented at places. We assume from the reader an in depth knowledge of the biomarker causal inference field and then follow the structure of many other papers published in this field (which is not necessarily a good thing per se). Still, we appreciate the reviewer's point and added in several instances clear signposts to justify (and clarify) the flow of the paper. We did our best to address all concrete questions of the reviewer. Importantly, we added the paragraph below to the introduction section and several others to other

sections:

Such significant enrichment suggests that many SNP-trait associations could act through gene expression (i.e. $\text{SNP} \rightarrow \text{Gene expression} \rightarrow \text{trait}$). [...] For this reason, we rather chose to apply a Mendelian randomization (MR) approach to estimate the causal effect of gene expression on complex traits. [...]

The manuscript is organized as follows. First, we performed an extensive simulation study to confirm that under our model setting, the method controls type I error rate and achieves superior RMSE compared to standard approaches. We then applied our method to the largest publicly available GWAS summary statistics (based on sample sizes ranging from 20,883 to 339,224 individuals) and combined them with eQTL data from GTEx (Genotype Tissue Expression Project)[29] and the eQTLGen Consortium (n=31,684) [30] to provide an atlas of putatively functionally relevant genes for 43 complex human traits. As there are only sporadic examples of causal gene-disease links, we used the following proxies to gold standard gene-disease links and tested whether the TWMR results are meaningful and confirm previous knowledge: (a) experimentally established causal links (e.g. *SORT1* with LDL in liver [31]); (b) gene-disease links based on the OMIM database; (c) genes falling into an association region identified by GWAS, but only in larger sample size. Finally, we carried out several follow-up analyses to make biological inferences.

Currently, it reads like the authors tried some things, and these are their observations - like a lab notebook rather than a hypothesis-led piece of science. There's other aspects of the analysis that again are described on the fly rather than in a principled way. The simulation - what specific properties of the method are the authors trying to show in the simulation study? There's currently no motivation for this.

We performed simulation analyses to demonstrate the advantage of our multi-exposure approach in settings relevant for expression-disease causal analysis.

Since one of the novel details of our implementation is the simultaneous inclusion of multiple genes in the MR model, we think it is important to show the improvement in precision of causal effect estimation given by such approach.

We clarified this point in the text:

To demonstrate the advantage of our multi-exposure approach, we performed simulation analyses in settings relevant for expression-disease causal analysis. In particular, we demonstrated that if a subset of SNPs affect more than one gene at a locus, the multi-gene approach provides a more precise estimation of the causal effect of the gene expression on the phenotypes: the root mean squared error (RMSE) in the multi-gene approach is >2-fold lower than in the single gene approach in all the simulations performed varying the degree of pleiotropy, number of genes and SNPs included in the model.

The section on "trait correlation" - there's no mention of this in the introduction or the methods. It may be a worthwhile thing to do, but it gives the impression that the authors did this because they could, rather than having a prior motivation for doing this.

As reported in Bulik-Sullivan et al (<https://www.nature.com/articles/ng.3406>), many traits share genetic effects.

In our analyses we want to see how much of the genetic overlap goes through gene expression, which is a biologically intriguing question. We added a sentence at the beginning of the paragraph:

Exploring the relationships among complex traits and diseases can provide useful etiological insights and help prioritize likely causal relationships. A cross-trait LD Score regression method [52] was used to evaluate the genome-wide genetic correlation between complex traits. To possibly understand the biological mechanism of the shared genetic architecture we estimated the proportion of such genetic correlation channeled through the transcriptome program. For this, we computed the correlation (ρ^E) between the causal effect estimates of the gene expression (or equivalently the Z-scores from our MR analysis) across a subset of 2,974 independent genes (including those that were not significant for any trait; see Methods).

Similarly for the section on "tissue-specific effects" - where did that come from?

Ongen et al (<https://www.nature.com/articles/ng.3981>) showed that whole blood is not always the right tissue to find causal genes for complex traits. Since GTEx is providing eQTLs data for 48 tissues, we extended our analyses to tissue-specific data. As reported in the text, these analyses allow us to find causal genes that we would have missed if our analyses would have been limited to whole blood eQTLs (*SORT1* associated with LDL in liver is a key example, where other tissues point to *PSRC1* as causal tissue and only liver points to *SORT1*).

This fact is mentioned at the beginning of the "Tissue-specific effects" section of the previously submitted manuscript:

Since many traits manifest themselves only in certain tissues, it is important to integrate data from the tissue of interest for the studied phenotype when trying to interpret GWAS results using gene expression as an intermediate phenotype.

Similarly for the part on discovering new loci - I thought the point of the method was to link gene expression to phenotypes?

Correct, the aim of the method is to link genes to phenotypes and better illuminate the genetic basis of complex traits. Looking at the results, we observed that many associations were new (i.e. no SNPs in that region was significantly associated with the trait in the previous GWAS). We think that it is crucial to show that, not only integration of eQTLs and GWAS data allows to prioritize genes in already known associated region but that our approach also has more power to unravel new associated regions missed by previous GWAS.

There even appears to be a new analysis introduced in the discussion: "For example, we

interrogate the 71 loci...".

In short, it would really help if the introduction had a paragraph saying: "In this paper, we will do X, Y and Z" and described the motivation for doing X, Y and Z, the data to be used, the hypothesis they are testing, what they expect to see, and so on. Currently, I find it hard to follow what analyses have been performed, I don't get why the different analyses are performed, what they are showing or how I'm supposed to know if they were successful or not.

As stated above, we modified the manuscript to better explain the logic of the paper. We hope that this reviewer will find this revised version easier to read.

Figure 1 is really helpful for understanding how the method was performed, and how the authors selected which genes to include in the analysis. How were the authors certain that they included all relevant genes in each analysis? Also, there appears to be some confusion about the pruning threshold (LD<0.1 here, $r^2 < 0.4$ in the methods section).

We have two different pruning threshold. We pruned the SNPs using 0.1 as threshold for LD, and we pruned the genes using 0.4 as threshold for r^2 .

This is clearly mentioned in the Methods section:

To avoid numerical instability in our multiple regression model, we pruned SNPs that are in high LD ($r^2 > 0.1$) (Fig. 1a). [...]

To avoid numerical instability caused by near-colinearity in our multiple regression model and making choices between co-regulated genes, we removed one gene from each pair of genes with $r^2 \geq 0.4$. The correlation r^2 was estimated as Pearson's correlation between the Z-scores of the shared, independent eQTLs.

"One way to guard against the violation of the third assumption is to use as many IVs possible, as the pleiotropic effect of each marker will cancel each other out under the INSIDE assumption (instrument strength independent of the strength of the pleiotropy)." This is completely untrue. There's no reason why pleiotropic effects would cancel out, even under the InSIDE assumption (pleiotropic effects can be independent and not average to zero). And this is not how assumptions work - you cannot invoke an assumption to make a point: it's like saying that A is true because I have assumed that A is true. The argument is circular.

We agree that this was awkwardly stated. What we meant is that using multiple instruments allows us to replace the often-violated third assumption with the weaker INSIDE assumption. Then, using multiple exposures could further weaken the INSIDE assumption. We have thus reworded the Results section ("Overview of the approach" subsection) as follows:

Including other SNPs as instruments allows us to replace the third MR assumption with the weaker INSIDE (INstrument Strength Independent of Direct Effect)

assumption. When the INSIDE assumption is violated, but entirely due to another genes' expression being the confounders of the (primary gene) expression-trait correlation, including the confounder genes as an additional exposures in a multivariable MR can resolve the problem and yield unbiased causal effect estimates.

And the Methods section as well:

Most probably the third assumption is almost always violated in practice, but can be replaced by the weaker INSIDE assumption (instrument strength independent of the strength of the pleiotropy) when multiple independent instruments are available.

Reviewers' Comments:

Reviewer #3:

Remarks to the Author:

The manuscript by Porcu et al applies established multivariable MR methodology to gene expression traits as exposures to increase power to detect gene/trait associations and address pleiotropy. The application of this approach is interesting and the authors provide good evidence of its value, although it is interesting that they don't also emphasise the practical benefit of highlighting causal genes (rather than just SNPs).

The authors have provided a detailed response to previous reviewers comments, and in most (but not all) cases have modified the manuscript to address the issues raised. I have no additional challenges on those points, but I do have some additional comments related to the key message of the manuscript, which I feel need to be addressed given the importance placed on this.

Major comments

- The section on new trait-associated genes helps underpin the core finding that "Our advanced Mendelian Randomization unlocks hidden value from published GWAS through higher power in detecting associations." This section relies heavily on cross-referencing to Mendelian traits (using OMIM), but without formal enrichment analysis to provide strength of evidence of this "validation". For example, given that there are over 3000 genes in OMIM with phenotypes relating to cognitive impairment (or similar), finding a relevant validation for a putative "educational attainment" locus could well be down to chance. The strength of evidence for these validations should be much better quantified if this section is to be retained.
- In the same section on new trait-associated genes, the authors state that 36% gene-trait associations have been "missed by previous GWASs". This is not very good wording, since the GWAS likely evaluated those associations, but filtered them out of the "top hits" based purely on p-value. In addition to changing the wording to state that these were not "prioritised" by previous GWAS, it would be helpful to evaluate whether those 36% include regions with signals evident at a more relaxed p-value threshold in GWAS - ie an analysis similar to that in the previous section (MR improves GWAS power to detect associated loci), but relaxing the p-value threshold instead of increasing sample size.
- The analysis in the previous section (MR improves GWAS power to detect associated loci) is definitely helpful in establishing that larger sample sizes validate the MR approach performed on smaller subsets. However, the 500kb threshold seems quite arbitrary, and it would be useful to have an indication of how this was selected.

Minor comments

- InSIDE assumption has a lower-case "n"

Reviewers' comments:

Reviewer #3 (Remarks to the Author):

The manuscript by Porcu et al applies established multivariable MR methodology to gene expression traits as exposures to increase power to detect gene/trait associations and address pleiotropy. The application of this approach is interesting and the authors provide good evidence of its value, although it is interesting that they don't also emphasise the practical benefit of highlighting causal genes (rather than just SNPs).

The authors have provided a detailed response to previous reviewers comments, and in most (but not all) cases have modified the manuscript to address the issues raised. I have no additional challenges on those points, but I do have some additional comments related to the key message of the manuscript, which I feel need to be addressed given the importance placed on this.

We thank the reviewer for his/her praise. We will do our best to address his/her comments

Major comments

- The section on new trait-associated genes helps underpin the core finding that "Our advanced Mendelian Randomization unlocks hidden value from published GWAS through higher power in detecting associations." This section relies heavily on cross-referencing to Mendelian traits (using OMIM), but without formal enrichment analysis to provide strength of evidence of this "validation". For example, given that there are over 3000 genes in OMIM with phenotypes relating to cognitive impairment (or similar), finding a relevant validation for a putative "educational attainment" locus could well be down to chance. The strength of evidence for these validations should be much better quantified if this section is to be retained.

We agree with the reviewer and we quantified the enrichment of OMIM genes in our TWMR-prioritized genes. We extracted from OMIM 1,026 genes associated with "cognitive impairment", "developmental delay", "intellectual disability", "intellectual impairment", "mental retardation", "mentally retarded", "cognitive delay" or "impaired intellectual development" and mapping to autosomes. We further restricted our search to genes with known sequence and phenotype. 669 of them were tested by TWMR and 9 were significantly associated with educational attainment. This resulted in a significant enrichment of 2.6-fold ($P=0.005$). Since in the section we also referred to significant genes found for height and total cholesterol, we extended the same analyses to these traits using lists of genes involved in abnormal skeletal growth syndrome and hypercholesterolemia, respectively.

We modified the manuscript accordingly, it now reads:

Furthermore, we prioritized genes in regions already known to be associated with complex traits. [...]

To test whether our putative causal genes are functionally relevant, we overlapped the genes significantly associated with height, educational attainment and total cholesterol with genes assembled from the Online Mendelian Inheritance in Man (OMIM) database involved in abnormal skeletal growth syndrome [43], cognitive impairment and hypercholesterolemia, respectively.

While we observed only a trend for enrichment for height (1.3-fold, $P > 0.05$) and total cholesterol (3.7-fold, $P > 0.05$), we found a significant enrichment for educational attainment (2.6-fold, $P = 0.005$) providing additional supporting evidence for our prioritized genes.

- In the same section on new trait-associated genes, the authors state that 36% gene-trait associations have been "missed by previous GWASs". This is not very good wording, since the GWAS likely evaluated those associations, but filtered them out of the "top hits" based purely on p-value. In addition to changing the wording to state that these were not "prioritised" by previous GWAS, it would be helpful to evaluate whether those 36% include regions with signals evident at a more relaxed p-value threshold in GWAS - ie an analysis similar to that in the previous section (MR improves GWAS power to detect associated loci), but relaxing the p-value threshold instead of increasing sample size.

We agree with this comment and changed the text accordingly.

Of these gene-trait associations, 36% (1,399) were not prioritized by previous GWASs, as no SNP reached genome-wide significance level within the gene +/- 500kb (Supplementary Fig. 10). Of note, 27% (1,068/3,913) were missed by GWASs even when using a less stringent threshold (i.e. $P_{\text{GWAS}} < 1 \times 10^{-6}$).

Please find below a plot where we show how the percentage of associations missed by GWAS decreases using more relaxed p-value thresholds. We would prefer not to include this figure in the paper since it is not more informative than Supplementary Figure 10.

- The analysis in the previous section (MR improves GWAS power to detect associated loci) is definitely helpful in establishing that larger sample sizes validate the MR approach performed on smaller subsets. However, the 500kb threshold seems quite arbitrary, and it would be useful to have an indication of how this was selected.

We thank the reviewer for the opportunity to clarify this point. We elected to use a 500kb threshold as it has been shown that cis-eQTLs rarely have any effect on expression levels of genes mapping at distance greater than 500kb (Marbach et al (Nat Methods 2016), supplementary Figure 1d <https://www.nature.com/articles/nmeth.3799#supplementary-information>). Selection of this threshold ensures that if we call a gene “missed by GWAS” it is highly improbable that a GWAS significant SNP could be linked (through eQTL) to the discovered TWMR gene (Zhu et al (Nat Gen 2016) <https://www.nature.com/articles/ng.3538>, Mancuso et al (AJHG 2017) [https://www.cell.com/ajhg/fulltext/S0002-9297\(17\)30032-0](https://www.cell.com/ajhg/fulltext/S0002-9297(17)30032-0)).

Minor comments

- InSIDE assumption has a lower-case "n"

We corrected this mistake in the text